# Environmental Impact Assessment of Solid Oxide Fuel Cell Power Generation System Based on Life Cycle Assessment—A Case Study in China

**Yilin Shen** [1,2], **Yantao Yang** [1,2], **Lei Song** [1,2] and **Tingzhou Lei** [1,2,*]

1    Institute of Urban & Rural Mining, Changzhou University, Changzhou 213164, China;
     syorlin11@163.com (Y.S.); yyt@cczu.edu.cn (Y.Y.); s22020817013@smail.cczu.edu.cn (L.S.)
2    National-Local Joint Engineering Research Center of Biomass Refining and High-Quality Utilization,
     Changzhou University, Changzhou 213164, China
*    Correspondence: china_newenergy@163.com; Tel.: +86-138-3716-6683

**Abstract:** To progress towards the "dual carbon" goal and reduce the cost and increase the efficiency of solid oxide fuel cells, this study conducts a full life cycle analysis of solid oxide fuel cells, in which the environmental impact caused by the operating devices' manufacturing, fuel gas catalyst reforming, single-cell manufacturing, cell stack manufacturing, and energy consumption and emissions are systematically analysed. In this study, we establish an assessment model for solid oxide fuel cells by using the cut-off criterion. The results show that 96.5% of the global warming potential in the use of solid oxide fuel cells comes from the stack operating subsystem. The stack manufacturing subsystem, operating device manufacturing subsystem, and waste stack processing subsystem all contribute greatly to acidification, accounting for 32.89%, 44%, and 35.82% of the total acidification, respectively. These three subsystems also contribute significantly to eutrophication, contributing 23.11%, 22.03%, and 42.15%, respectively. Compared with traditional thermal power generation systems, solid oxide fuel cell power generation systems have slightly higher overall environmental benefits, and the reductions in greenhouse gas emissions and acidification potential reach 6.22% and 18.52%, respectively. The research results have guiding significance and reference value for subsequent energy-saving and emission reduction design and improvement efforts for solid oxide fuel cells.

**Keywords:** solid oxide fuel cells; life cycle assessment; environmental impact; greenhouse gas; new energy technology

## 1. Introduction

Fossil energy sources, with their high carbon emissions, have long dominated the global energy supply. Given the environmental crisis and the rapid depletion of non-renewable resources, the world is facing an energy emergency, especially concerning electricity. Natural-gas-fuelled solid oxygen fuel cells (SOFCs) are power generation systems that use hydrocarbon reforming to produce hydrogen, mainly produced via a gas-phase reaction between methane and water vapour. Life cycle assessment (LCA), an objective process of evaluating the environmental impact and energy consumption of products, processes and activities, is the most frequently used method to measure the environmental impact of using SOFC power generation systems [1]. In this method, the global warming potential (GWP), acidification potential (AP), and eutrophication potential (EP) are common evaluation indicators [2–4]. Taking an SOFC power generation system as the study object, this study aims to use LCA to assess the energy consumption and environmental emissions of each SOFC subsystem quantitatively, evaluate the environmental impact potential of the overall system, and compare the results with traditional thermal power generation systems. Furthermore, this study identifies the advantages of SOFC power generation systems

and proposes improvement and optimization solutions. The results of this study provide valuable theoretical support for promoting the sustainable development and commercial application of SOFC power generation plants.

## 2. Materials and Methods

The methodological framework of an LCA, as defined in the ISO 14044:2006 [1] standard, includes four main phases:

- Definition of goal and scope, where the aim of the study is delineated, its breadth and depth are established, and functional units and system boundaries are set;
- Life cycle inventory (LCI), where data collection is performed, including calculation and allocation procedures;
- Life cycle inventory assessment (LCIA), where the potential environmental effects related to the results of the inventory analysis are evaluated;
- Interpretation, where the results of the LCIA are analysed and combined in relation to the goal and scope of the research.
- The application of LCA methods to SOFC systems has matured in international studies; in this study, we use a local case study in China for a brief analysis.

### 2.1. Goal and Scope definition

An SOFC power generation system with a rated power of 2 kW manufactured by a Chinese New Energy company was used as the object of this study, and the system boundaries were determined via on-site research, literature reviews [5,6], and consultations with experts (Figure 1).

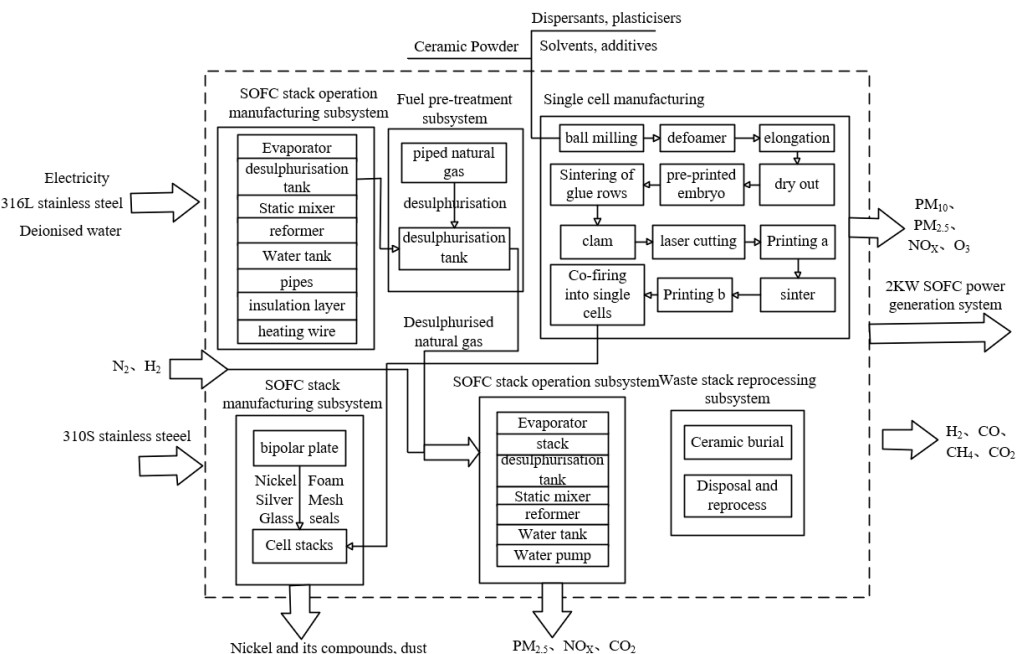

**Figure 1.** Boundary of SOFC power generation system.

Plant layout and Flow Process

The production process of stacks was assessed using on-site research. The stack operation manufacturing subsystem considers the design and manufacture of the insulation layer, evaporator, desulphurization tank, static mixer, reformer, water tank, and pipelines of the SOFC system. The insulation layer is made of aluminosilicate materials, the water tank and pipelines are made of 316 L stainless steel, and the other devices are made of 310 S stainless steel. The organic and inorganic sulphur content of pipeline natural gas in the Jiangsu region is about 2 ppm, and the natural-gas-fuelled SOFC power generation system needs to

reduce the sulphur content to a few ppb; otherwise, it will poison both the reforming and electrochemical activity of the anode [7]. Therefore, a fuel pretreatment subsystem is added to the whole manufacturing process. The single-cell manufacturing subsystem is the core stage of the whole system, including processes of powder weighing, ball milling, casting, printing, cutting, and sintering. For example, to manufacture 20 single cells, ceramic powder, a dispersant, additives, and other materials are required, totalling 2.15 kg. The stack manufacturing subsystem has to consider the design and manufacturing of bipolar plates and the assembly and sealing of the single cells. Moreover, the stack operation subsystem has to consider the input of two protective gases ($N_2$ and $H_2$). A cell stack reprocessing and treatment system has not been established in the actual process on site, so ceramic burial and some scrap stacking of the waste stacks were considered in conjunction with the large body of literature. The full life cycle analysis of the integrated system was based on the production of a 2 kW stack system (Figure 2). The inputs to the integrated system include hydropower for production, and the outputs include environmental pollutants discharged from subsystems and the used products. Due to the lack of data obtained, environmental impacts caused by the construction of units, the transport energy consumed in transport of the subsystems, etc., were ignored.

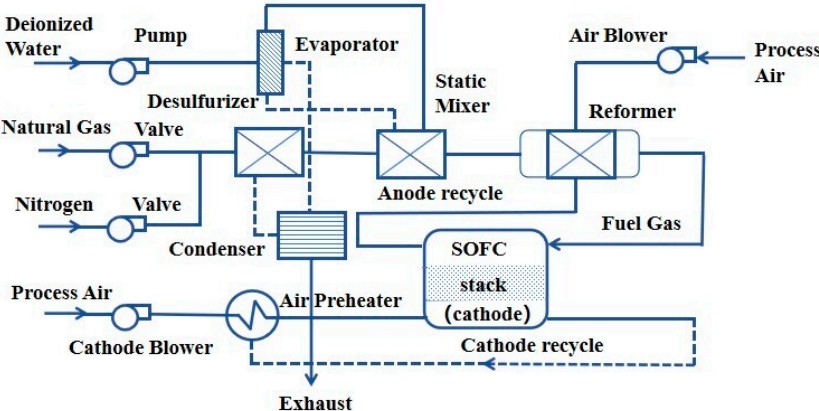

**Figure 2.** Process flow diagram of the SOFC system.

## 2.2. Inventory Analysis

Through on-site research and consultation with experts on SOFC power generation systems, combined with environmental emission factors and energy consumption factors, the inventories of the stack operating device manufacturing subsystem, fuel pre-treatment subsystem, single-cell manufacturing subsystem, stack manufacturing subsystem, stack operation subsystem, and waste stack reprocessing subsystem were analysed separately.

### 2.2.1. SOFC Stack Operating Device Manufacturing Subsystem

According to the field study, the operating device manufacturing subsystem consumes 29.8 kg of 316 L stainless steel, 10.2 kg of 310 S stainless steel, and 15.3 kg of silicic acid; as the specific process in the manufacturing stage of the operating unit is no longer available, the environmental impact of the stainless steel was calculated under the welding conditions of the smoky rain model [8].

$$C_{O_3} = 0.000000282V_X^2 + 0.00109V_X \tag{1}$$

$$C_{NO_X} = 0.00000146V_X^2 + 0.000568V_X \tag{2}$$

$$C_{PM_{2.5}} = 0.000000163V_X^2 - 0.0000981V_X \tag{3}$$

$$C_{PM_{10}} = 0.000000135V_X^2 - 0.0000119V_X \tag{4}$$

where $C_{O_3}$, $C_{NO_X}$, $C_{PM_{2.5}}$, and $C_{PM_{10}}$ indicate the concentrations of ozone ($O_3$), nitrogen oxide ($NO_X$), particulate matter $\leq 2.5$ μm ($PM_{2.5}$), and particulate matter $\leq 10$ μm ($PM_{10}$) in 0.11 m$^3$ of space, respectively. $V_X$ is the arc covering area.

Table 1 lists the inputs and outputs of raw materials, energy, and resources of the SOFC stack operating device manufacturing subsystem.

**Table 1.** Analysis of stack operating device manufacturing subsystem list.

| Event | Unit | Mass |
|---|---|---|
| *Input* | | |
| 310 S stainless steel | kg | 10.2 |
| 316 L stainless steel | kg | 29.8 |
| Aluminosilicate | kg | 15.3 |
| *Output* | | |
| $O_3$ | g | 2.798 |
| $NO_x$ | g | 14.317 |
| $PM_{2.5}$ | g | 0.156 |
| $PM_{10}$ | g | 0.132 |

### 2.2.2. Fuel Pre-Treatment Subsystem

The power generation system is fuelled by pipeline natural gas, which is treated in a desulphurization unit and then passed into a static mixing tank in the form of methane ($CH_4$) with a concentration fraction of 99.9%. The gas is then mixed with water vapour for hydrocarbon reforming, with the following reforming reaction equation:

$$CH_4 + H_2O = 3H_2 + CO \tag{5}$$

The methane steam reforming process is a reversible adsorption reaction process in which carbon dioxide and other impurities are removed from the gas stream during the final variable pressure adsorption process, leaving essentially pure hydrogen and carbon monoxide [9].

Table 2 lists the inputs and outputs of raw materials, energy, and resources of the fuel pre-treatment subsystem.

**Table 2.** Analysis of fuel pre-treatment subsystem list.

| Event | Unit | Mass |
|---|---|---|
| *Input* | | |
| Piped natural gas | L | 2906.4 |
| Desulphurization | kg | 1 |
| Deionized water | m$^3$ | 2906.4 |

### 2.2.3. Single-Cell Manufacturing Subsystem

The single-cell manufacturing subsystem involves more complicated processes, including powder weighing, slurry ball milling, casting and forming, gluing and smoothing, printing and sintering, and cutting (Figure 3). Referring to the EIA report prepared by the company in 2019, it can be seen that nickel and its compounds and VOCS will be generated from the processes of pulping, sintering, and silk screen printing during the manufacturing of single cells, amounting to outputs of 5.22 kg·a$^{-1}$ and 57.8 kg·a$^{-1}$, respectively, after the capture measurement. The sintered half-cells require secondary processing to reduce their original size, and the environmental impact caused by the cutting process is negligible here.

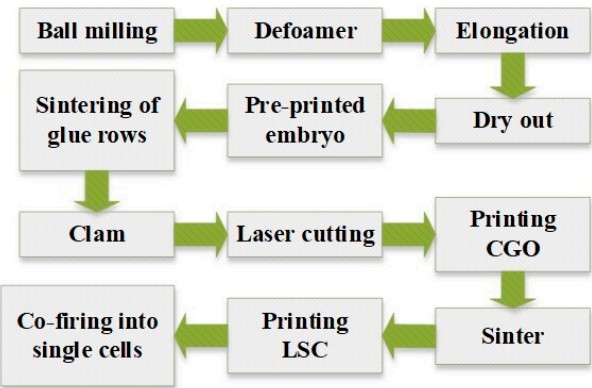

**Figure 3.** Single-cell manufacturing subsystem process.

The power consumption in the single-cell manufacturing stage depends mainly on the power consumption of large equipment such as vacuum pumps, ball mills, casting machines, printing presses, and chamber furnaces. According to the on-site research, the specific power consumption of producing a batch of single-cell wafers can be calculated according to the following steps:

① Use a tank mill with a rated power of 2.2 kW to grind the weighed slurry;
② Use a vacuum pump with a rated power of 0.18 kW to defoam the slurry after the treatment in ①;
③ Use a casting machine with a rated power of 60 kW to cast the slurry treated in ②;
④ Print the cells with a press rated at 1.5 kW;
⑤ Roast the battery sheet using a chamber furnace with rated powers of 26 kW, 26 kW, 21.4 kW, and 40 kW;
⑥ Process the semi-finished cell wafers into fixed sizes.

The power consumption of the 2 kW SOFC power generation system is 10.634 MWh.

The average generation efficiency of conventional power plants in China is 37%, and the transmission and distribution efficiency of the power grid is 93%. The energy consumption of the production process is calculated as follows:

$$Q_c = \frac{E_e \times 3.6}{\eta_e \eta_{grid}} \tag{6}$$

where $\eta_e$ is the average power generation efficiency of the power plant, %; $\eta_{grid}$ is the transmission and distribution efficiency of the power grid, %; $E_e$ is the energy consumed, kwh; and QC is the heat converted into raw coal, MJ.

Table 3 lists the inputs and outputs of raw materials, energy, and resources of the single-cell manufacturing subsystem. Some data were referenced from the literature and filled out based on actual situations [10–18].

**Table 3.** Analysis of single-cell manufacturing subsystem list.

| Event | Unit | Mass |
|---|---|---|
| *Input* | | |
| Yttrium oxide-stabilized zirconia (anode) | g | 120 |
| Yttrium oxide-stabilized zirconia (sintered) | g | 2388 |
| Nickel oxide | g | 2760 |
| Dimethyl phthalate | g | 264 |
| Dimethyl phthalate (solvent) | mL | 21.6 |
| Polyethylene glycol | g | 396 |
| Polyvinyl butyral | g | 521.4 |
| Triethanolamine | g | 9.6 |
| Butanone | g | 1956 |

**Table 3.** *Cont.*

| Event | Unit | Mass |
|-------|------|------|
| *Input* | | |
| Anhydrous ethanol | mL | 4164 |
| Corn starch | g | 624 |
| Citrate | g | 27.5 |
| Lanthanum nitrate | g | 18.33 |
| Strontium nitrate | g | 18.33 |
| Cobalt nitrate | g | 18.33 |
| Electrical | kw·h | 10,633.77 |
| *Output* | | |
| Coal consumption | $10^6$ MJ | 0.111 |
| Complicated cells | Piece | 60 |
| Nickel and its compounds | $kg·a^{-1}$ | 5.22 |
| VOCS | $kg·a^{-1}$ | 57.8 |

2.2.4. SOFC Stack Manufacturing Subsystem

The main processes in the stack manufacturing subsystem are the manufacture of bipolar plates and glass seals and the assembly of cells (Figure 4). The object of this evaluation is the rated power of the 2 kw SOFC power generation system, with 60 single-cells, which requires 60.6 kg of 310 S stainless steel, 40 g of glass seals, 600 g of silver mesh, and 160 g of nickel foam.

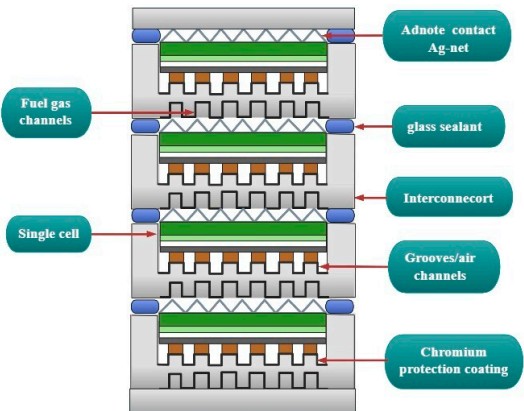

**Figure 4.** Structure of the cell stack.

Table 4 lists the inputs and outputs of raw materials, energy, and resources of the SOFC stack manufacturing subsystem.

**Table 4.** Analysis of stack manufacturing subsystem list.

| Event | Unit | Mass |
|-------|------|------|
| *Input* | | |
| 310 S stainless steel | kg | 60.6 |
| Nickel foam | kg | 0.16 |
| Silver network | kg | 0.6 |
| Glass seals | kg | 0.04 |
| Single cell | Piece | 60 |
| *Output* | | |
| $CO_2$ | kg | - |
| $NO_X$ | kg | - |
| $PM_{2.5}$ | kg | - |
| Complicated stack | Classifier | 1 |

### 2.2.5. SOFC Stack Operation Subsystems

The cell stack operation subsystem is actually the core stage of the 2 kW SOFC power generation system. This subsystem involves a number of unit components, including evaporators, desulfurization tanks, static mixers, reformers, pipes, aluminosilicate insulation materials and appropriate booster pumps, flow meters, heat exchangers, etc. (Figure 2). During the operation of the cell stack, a large amount of tail gas is generated from both the cathode and anode. The anode gas undergoes an electrochemical reaction, and in addition to generating a large amount of $H_2O$ and $CO_2$, residual, unreacted combustible gases such as $H_2$ and CO will persist, as well as a small amount of $N_2$ and $O_2$. After collecting and analysing the anode's exhaust gas (92.54%), we determine that the concentration ratio of $CO_2$, $H_2$, CO, $CH_4$, $N_2$, and $O_2$ in the anode exhaust gas (92.54%) is 2.03:49.66:11.7:21.36:5.05:2.74. Table 5 lists the inputs and outputs of raw materials, energy, and resources of the SOFC stack operation subsystem.

**Table 5.** Analysis of stack operation subsystem list (based on 4000 h system operation).

| Event | Unit | Mass |
|---|---|---|
| *Input* | | |
| Desulphurized natural gas | $m^3$ | 2906.4 |
| After burner | $m^3$ | 11,181.6 |
| Cathode fans | $m^3$ | 33,544.8 |
| Ni-$Al_2O_3$ catalysts | $m^3$ | 0.06 |
| Deionized water | $m^3$ | 2906.4 |
| *Output* | | |
| $CO_2$ | kg | 171.245 |
| CO | kg | 581.28 |
| $H_2$ | kg | 118.742 |
| $CH_4$ | kg | 388.897 |

### 2.3. Impact Evaluation

The main types of environmental impacts considered in this study (Table 6) were the global warming potential (GWP), acidification potential (AP), eutrophication potential (EP), fine particulate matter formation (PMF), ozone depletion potential (ODP), and non-renewable energy consumption (NREC). The six types of environmental impacts were evaluated and analysed using the Characterization, Weighted and Normalization Method, supplemented by GaBi9 software, to derive the environmental impact index. The potential environmental impact and weighting factors (Table 7) were referenced from Wang S-B (2004) and Fang Z-C (2022)'s research [19,20].

**Table 6.** Environmental impact types of SOFC stack integrated system.

| Type of Impact | Influencing Substances | Equivalent Factor |
|---|---|---|
| GWP | $CO_2$ | 1 |
| | CO | 2 |
| | $CH_4$ | 25 |
| | $N_2O$ | 265 |
| AP | $SO_2$ | 1 |
| | $NO_X$ | 0.7 |
| | $NO_2$ | 1.15 |
| EP | $PO_4$ | 1 |
| | $NO_X$ | 0.1 |
| PMF | $PM_{2.5}$ | 1 |
| | Dust | 4 |
| ODP | $O_3$ | 1 |

**Table 7.** Global per-capita environmental impact potential and weighting factors.

| Typology | Unit | Per Capita Equivalent | Weighing Factor |
|:---:|:---:|:---:|:---:|
| GWP [1] | kg·a$^{-1}$ | 8700 | 0.27 |
| AP [2] | kg·a$^{-1}$ | 35 | 0.18 |
| EP [3] | kg·a$^{-1}$ | 59 | 0.088 |
| NREC | MJ·a$^{-1}$ | 56,877.88 | 0.15 |

[1] calculated as $CO_2$; [2] calculated as $SO_2$; [3] calculated as $PO_4$.

In this project, gasses whose emissions will produce global warming impacts are $CO_2$, CO, and $CH_4$; when calculating the GWP, $CO_2$ is generally selected as the reference gas, and the other substances are replaced by $CO_2$ in equal quantities for calculation. Gasses whose emissions influence the AP are $SO_2$, $NO_X$, and $NO_2$; when calculating AP, $SO_2$ is generally selected as the reference gas, and the other substances are replaced by $SO_2$ in equal quantities for calculation. Gaseous emissions that will have an impact on the EP are $NO_2$ and $NO_X$; $PO_4$ is generally selected as the reference for calculating EP, and the rest of the substances are replaced by $PO_4$ in equal quantities for the calculation. Gaseous emissions that will have an impact on the PMF are $PM_{2.5}$ and general dust, and $PM_{2.5}$ is generally selected as the reference for calculating the PMF. Finally, only the ODP generated by $O_3$ is taken into account.

Characterization, Normalization, and Weighting

The characterization is calculated as follows:

$$C_j = x_z \times X_j \tag{7}$$

where C is the characterization result; x is the pollutant emissions from the functional unit, kg; X is the equivalence coefficient; j is the type of environmental impact; and z is the attribution of different substances to the same type of environmental impact.

The normalized formula is calculated as follows:

$$N_j = C_j / S_j \tag{8}$$

where N is the characterization result, S is the baseline value, R is the environmental impact index, and j is the weighting factor.

## 3. Results and Discussion

### 3.1. Environmental Impact Potential

The environmental impact potential of each subsystem was obtained via the joint calculation of the equivalence coefficient method as well as CML2001 (Table 8). The results show that the cell stack operation subsystem phase emitted more $CO_2$, and its GWP reached 2.863 kg. AP (calculated using $SO_2$) and EP (calculated using $PO_4$) reached $1.002 \times 10^{-2}$ kg and $1.432 \times 10^{-3}$ kg, respectively, in the stage of the operating unit manufacturing subsystem due to the processing of a large amount of stainless steel. The value of the formation of fine particles in the stage of the single-cell manufacturing subsystem was 58.625 kg, and the value of the ozone depletion potential in the stage of the operating unit manufacturing subsystem, due to the welding of a large amount of stainless steel, was $2.798 \times 10^{-3}$ kg (Figure 5).

**Table 8.** Potential environmental impact of each system.

| Subsystems | GWP [1]/kg | AP [2]/kg | EP [3]/kg |
|---|---|---|---|
| SOFC stack operating device manufacturing | - | $1.002 \times 10^{-2}$ | $1.432 \times 10^{-3}$ |
| Fuel pre-treatment | - | - | - |
| Single-cell manufacturing | - | - | - |
| SOFC stack manufacturing | - | $1.435 \times 10^{-2}$ | $0.880 \times 10^{-3}$ |
| SOFC stack operation | 2.763 | - | - |
| Waste stack processing | 0.100 | $1.931 \times 10^{-2}$ | $1.684 \times 10^{-3}$ |
| Total | 2.863 | $4.368 \times 10^{-3}$ | $3.996 \times 10^{-3}$ |

| Subsystems | PMF [4]/kg | ODP [5]/kg | NREC [6]/MJ |
|---|---|---|---|
| SOFC stack operating device manufacturing | $0.156 \times 10^{-3}$ | $2.798 \times 10^{-3}$ | - |
| Fuel pre-treatment | - | - | - |
| Single-cell manufacturing | 12.84 | - | 111,000 |
| SOFC stack manufacturing | $0.736 \times 10^{-3}$ | - | - |
| SOFC stack operation | - | - | 111,780.14 |
| Waste stack processing | $1.942 \times 10^{-3}$ | - | - |
| Total | 12.855 | $2.798 \times 10^{-3}$ | 222,780.14 |

[1] calculated as $CO_2$; [2] calculated as $SO_2$; [3] calculated as $PO_4$; [4] calculated as $PM_{2.5}$; [5] calculated as $O_3$; and [6] calculated as coal consumption.

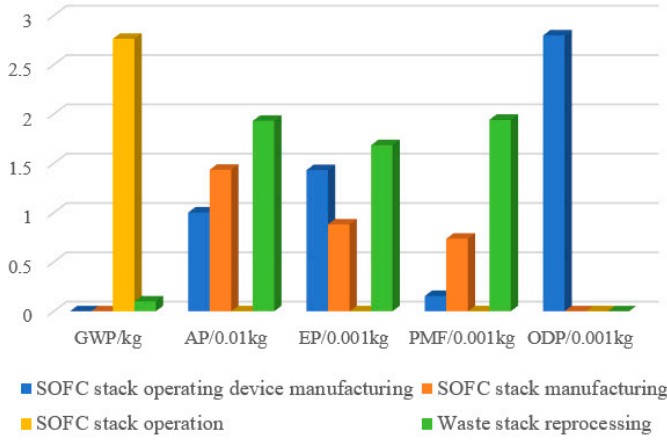

**Figure 5.** Calculation of system-equivalent environmental impact potential.

### 3.2. Analysis of Environmental Impact Contributions

According to the data in Table 8, the environmental impact index of each subsystem (Table 9) can be calculated using Formulas (9) and (10). Based on the data in Table 9, the percentage contribution of each subsystem for each of the five environmental impact types (Figure 6) was plotted and analysed.

**Table 9.** Environmental impact index of each system.

| Subsystems | GWP | AP | EP |
|---|---|---|---|
| SOFC stack operating device manufacturing | - | $0.52 \times 10^{-4}$ | $2.135 \times 10^{-6}$ |
| Fuel pre-treatment | - | - | - |
| Single-cell manufacturing | - | - | - |
| SOFC stack manufacturing | - | $0.74 \times 10^{-4}$ | $1.313 \times 10^{-6}$ |
| SOFC stack operation | $8.575 \times 10^{-5}$ | - | - |
| Waste stack processing | $3.103 \times 10^{-6}$ | $0.99 \times 10^{-4}$ | $2.512 \times 10^{-6}$ |
| Total | $8.885 \times 10^{-5}$ | $2.25 \times 10^{-4}$ | $5.960 \times 10^{-6}$ |

**Table 9.** *Cont.*

| Subsystems | PMF | ODP | Total (5 indices) |
|---|---|---|---|
| SOFC stack operating device manufacturing | $4.841 \times 10^{-9}$ | $8.683 \times 10^{-8}$ | $5.423 \times 10^{-5}$ |
| Fuel pre-treatment | - | - | - |
| Single-cell manufacturing | $3.986 \times 10^{-4}$ | - | $3.986 \times 10^{-4}$ |
| SOFC stack manufacturing | $2.284 \times 10^{-8}$ | - | $7.534 \times 10^{-5}$ |
| SOFC stack operation | - | - | $8.575 \times 10^{-5}$ |
| Waste stack processing | $6.018 \times 10^{-8}$ | | $1.047 \times 10^{-4}$ |
| Total | $3.987 \times 10^{-4}$ | $8.683 \times 10^{-8}$ | $7.186 \times 10^{-4}$ |

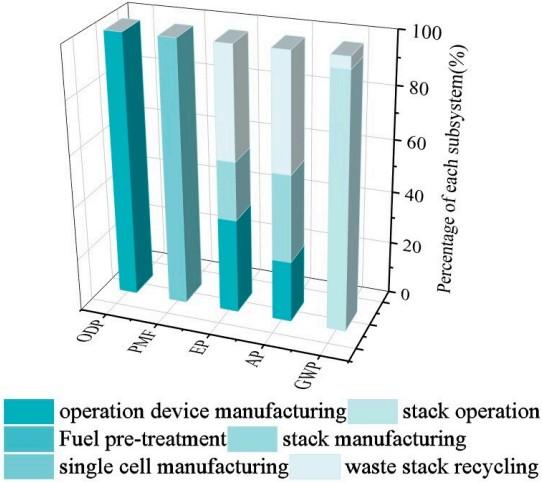

**Figure 6.** Proportion of different environmental impact indicators for each subsystem.

### 3.3. Global Warming and Non-Renewable Energy Consumption

The main contribution to the GWP in this SOFC power generation integrated system comes from the tail gas emissions of the SOFC stack operation subsystem (96.5%), which finding is in line with Carlo Strazza's study [2]; and the total $CH_4$ emissions caused by incomplete reforming reactions are 388.897 kg/4000 h, which occupy 87.94% of this phase (Figure 7). From the inventory analysis, it can be seen that in the SOFC stack operation subsystem, after desulphurization treatment and mixing the fuel gas with water vapour to generate hydrogen-rich gas to support the operation of the system, not only will a small portion of $CO_2$ and CO be discharged from the reaction, but part of the methane gas will be discharged as well, which can be recycled and reused as the stack anode's fuel gas in a later stage. Considering ceramic treatment, the waste stack processing subsystem will produce 0.1 kg of $CO_2$. Few of the remaining subsystems contribute to the GWP. The consumption of non-renewable energy in the SOFC system is mainly the electricity consumed in the single-cell manufacturing stage and fuel gas consumption in the operation stage. The equivalent coal consumption is $1.11 \times 10^6$ MJ and $1.1178 \times 10^6$ MJ, respectively.

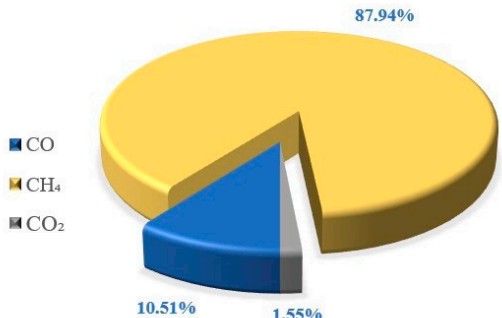

**Figure 7.** GWP map during operation.

### 3.4. Acidification and Eutrophication

Acidification and eutrophication of the SOFC system are concentrated in three phases: the SOFC operating device manufacturing subsystem, the SOFC stack manufacturing subsystem, and the waste stack processing subsystem. The SOFC operating device manufacturing subsystem and the SOFC stack manufacturing subsystem generate 23.11% and 32.89% of the total AP and 35.82% and 22.03% of the total EP, respectively. These contributions are due to the fact that stainless-steel materials emit large quantities of $SO_2$, $NO_X$, and other acidic gases during processing. Table 9 also shows that the AP and EP from the processing phase of waste stacks account for 44% and 42.15% of the total AP and total EP, respectively.

### 3.5. Fine Particulate Matter Formation and Ozone Depletion Potential

Concerning the environmental impact indicator of fine particle formation, 99.978% of fine particles are formed in the single-cell manufacturing stage. This is because the mixing and grinding of large quantities of ceramic powders and reagents in the manufacture of the cathode, anode, and electrolyte slurries invariably increase the density of dust in the slurry room. In addition, the vast majority of gases generated by the sintering of cells in the chamber furnace will be treated in an environmentally friendly manner, with the remainder being centrally discharged through the rooftop ventilation ducts. A small amount of particulate matter is also emitted to the surrounding environment due to stainless-steel processing during the manufacturing phase of the operating device and cell stack, accounting for 0.001% and 0.006% of the total particulate matter formation, respectively. Furthermore, an ozone depletion potential is present in the manufacturing phase of the operating device, which produces about $2.798 \times 10^{-3}$ kg of ozone gas.

### 3.6. Comparative Analysis of a Thermal Power System and the SOFC Power System

Based on the life cycle assessment of a biomass power generation conducted by Zhao H-Y on the output list of thermal power generation system producing $1 \times 10^4$ KWh of electricity [21], the percentages of its environmental impact indices after substitution calculation are listed in Table 10.

**Table 10.** Proportions of environmental impact indices.

| Event | | GWP | AP | EP |
|---|---|---|---|---|
| Thermal power system Zhao H-Y (2010) [21] | Mass | $1.006 \times 10^{-4}$ | $3.273 \times 10^{-4}$ | $1.063 \times 10^{-6}$ |
| Fuel pre-treatment | Proportions | 23.452% | 76.300% | 0.248% |
| SOFC power system | Mass | $8.885 \times 10^{-5}$ | $2.25 \times 10^{-4}$ | $5.960 \times 10^{-6}$ |
| SOFC stack manufacturing | Proportions | 27.782% | 70.354% | 1.864% |

As can be seen from Table 10, the overall environmental impact indicators of the SOFC power generation system are smaller than those of the thermal power generation system, and among the three environmental impact indicators (GWP, AP, and EP), AP is dominant. Table 10 shows that the thermal power generation system's AP accounted for 76.3% of the three impact indicators, while the SOFC power generation system's AP accounted for 70.354% of the three impact indicators. These results indicate that $NO_X$ gas emitted from either power generation system has a significantly higher environmental impact than the other gases. The SOFC power generation system studied here is still in the preliminary stages of research and testing, and the incomplete reforming reaction of the input desulphurized natural gas in the fuel pre-treatment subsystem has resulted in some $CH_4$ gas being discharged along with the gas stream during operation; this gas can be subsequently recycled and reused later to significantly reduce the GWP. The waste stack processing subsystem is not included in the discussion of environmental impact indicators in most SOFC life cycle assessments. In this paper, we use partial recycling and reprocessing and partial stacking procedures for waste products in our calculations, and

the results show that the operation device manufacturing stage and the waste processing stage produce nearly the same percentage of EPs. The LCA of the thermal power system does not take into account the waste treatment stage of raw materials and equipment and, therefore, has a higher EP only during the operation of the power plant.

As can be seen from Figure 8, the AP of the thermal power generation system during plant operation, which is 59.03%, is highest across all stages of the two systems, and the GWP of thermal power generation system during plant operation and raw material acquisition accounted for 52.90% of the total across all stages of the two systems. Meanwhile, the GWP of the SOFC system during the operation stage accounts for 45.30% of the total GWP across both systems. According to the above analysis, the traditional thermal power generation system using non-renewable coal as a raw material emits a large amount of $CO_2$, $SO_2$, and other greenhouse gases in the tail gas, consumes more non-renewable resources, and pollutes the atmosphere, whereas solid oxide fuel cells use cleaner natural gas as the raw material, and the combustion products are $CO_2$ and $H_2O$. Among these gases, $CO_2$ can be recycled into the reforming system and converted into CO for reuse; in addition, the exhaust emissions will be lower after the optimization of this system. The research results are significant because they can guide energy conservation and emission reduction efforts in the future. In view of this study, the authors suggest that more environmentally friendly hydrogen energy can be used as the fuel in the SOFC system to reduce carbon emissions to a minimum. In addition, the anode's electrode material and structure can be adjusted; not only can a more efficient and stable electrode improve the efficiency of the cells, but it can also reduce the loss of materials and improve the regeneration and replacement cycle, reducing the amount of waste and pollution. Finally, the authors suggest improving the heat conversion efficiency, reducing the cost, and improving the market competitiveness of solid oxide fuel cells.

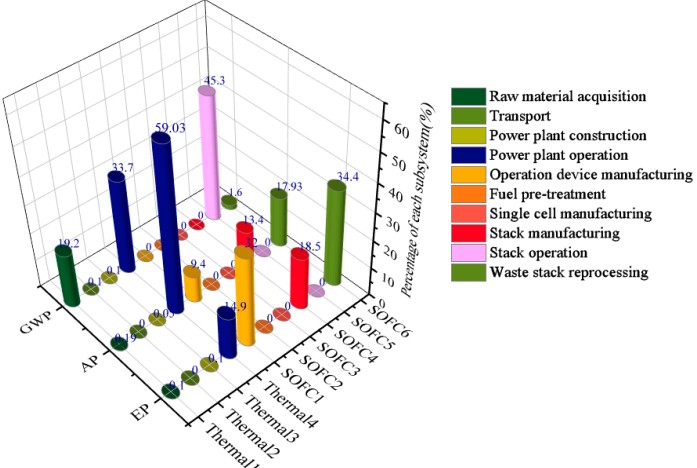

**Figure 8.** The proportion of three environmental impact indicators in each stage.

## 4. Uncertainties

Life cycle assessments of the environmental impact of SOFC power systems are a relatively comprehensive method for analysing distributed power generation systems, which are still in the developmental stages in China. Considering the complexity of the research content and the research object, the influencing factors used may lead to inconsistencies in the results with previous reports, as described below.

First, the activity data and emission factors used in the assessment process are partly derived from relevant studies, and the data concerning inputs and outputs of substances and energy at various subsystems are very complex and numerous, with a certain degree of uncertainty. Second, current research in this area has not identified a unified assessment model for SOFC systems using different fuels and different preparation processes, so the choice of system boundaries is subjective. Therefore, the final life cycle assessment of the

SOFC system may not be representative. As scholars continue research in this field, the relevant assessment models will become more mature and more precise. Despite these uncertainties in the assessment, our research, which analyses the environmental impacts caused by SOFC power generation systems using the LCA methodology, provides good insights for the design of subsequent systems to reduce emissions.

**5. Conclusions**

The following conclusions were obtained from the above evaluation:

(1) In SOFC power generation systems, the SOFC stack operation subsystem has the highest global warming potential, accounting for 96.5% of the GWP of the entire system. The formation of fine particulate matter is concentrated in the single-cell manufacturing subsystem, with an environmental impact index of $3.986 \times 10^{-4}$, and the ozone depletion potential is concentrated in the SOFC operating device manufacturing phase, with an environmental impact index of $8.683 \times 10^{-8}$.

(2) In the SOFC power generation system, the SOFC operating device manufacturing subsystem accounts for 23.11% of the acidification potential and 35.82% of the eutrophication potential, while the SOFC stack manufacturing subsystem accounts for 32.89% of the acidification potential and 22.03% of the eutrophication potential.

(3) Thermal power generation systems have high acidification potential and eutrophication potential indices during their operational phases. The reductions in greenhouse gas emissions and acidification potential of the SOFC power generation system compared to the thermal system reached 6.22% and 18.52%, respectively.

**Author Contributions:** Y.S.: conceptualization, writing—original draft, and investigation; Y.Y.: writing—review and editing and project administration; L.S.: formal analysis and software; T.L.: methodology, resources, and supervision. All authors have read and agreed to the published version of the manuscript.

**Funding:** This research received no external funding.

**Institutional Review Board Statement:** Not applicable.

**Informed Consent Statement:** Not applicable.

**Data Availability Statement:** Data are contained within the article.

**Acknowledgments:** The research has been supported by a Chinese New Energy company, whose project is classified. The content of this research reflects solely the authors' view, and the authors sincerely wish to here thank all the partners of the project.

**Conflicts of Interest:** The authors declare no conflicts of interest.

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
