# Peer review of "Environmental Impact Assessment of Solid Oxide Fuel Cell Power Generation System Based on Life Cycle Assessment—A Case Study in China"

_sustainability, doi:10.3390/su16093863_

Round 1

Reviewer 1 Report

Comments and Suggestions for Authors

Comment 1:Page 2 Line 44. In view of the shortcomings of SOFC power generation system, the author lacks personal innovative opinions. Please provide a feasible solution.

Comments 2:Page 3 Line73 What is the basis for the investigation of organic and inorganic sulfur in pipeline natural gas in Jiangsu. Please provide the cited literatures?

Comments 3:In Figure 3,The author’s description of the title does not match the content of the picture, please check the correct.

Comments 4:Page 13 Line301 .The author claimed three environmental impact indicators of GWP, AP, and EP about thermal power generation system and SOFC power generation system, which should be used to explain the result in Figure 8 instead of Figure 4, and the author should elaborate the comparative analysis of each environmental impact index of the two systems.

Comment 5:Page 14 Line321 The picture’s horizontal coordinates are not clear, please redraw it.

Comment 6:Please carefully correct some grammar errors.

Comments on the Quality of English Language

Minor editing of English language is required.

Reviewer 2 Report

Comments and Suggestions for Authors

This paper provides a comprehensive life cycle analysis of solid oxide fuel cells, offering valuable insights into their environmental impact across various stages of production and operation. The findings underscore the importance of addressing emissions from stack operating and manufacturing subsystems, as well as waste stack recycling, to mitigate environmental burdens. Despite slight increases in overall environmental benefits compared to traditional thermal power generation, the study highlights areas for improvement in solid oxide fuel cell design and operation to enhance energy efficiency and emissions reduction. This manuscript can be recommended for publication with minor revision.

1.      How does the environmental impact of solid oxide fuel cell power generation compare to traditional thermal power generation?

2.      Prioritize research and development efforts aimed at optimizing manufacturing processes for solid oxide fuel cell stacks to reduce environmental burdens associated with production. This may include exploring alternative materials, refining manufacturing techniques, and adopting cleaner energy sources to mitigate acidification and eutrophication potentials.

3.      What were the reductions in greenhouse gas emissions and acidification potential associated with solid oxide fuel cell power generation?

4.      What implications do the research findings have for the design and improvement of solid oxide fuel cells?

5.      How can the results of this study guide future efforts in energy-saving and emission reduction strategies?

6.      In what specific ways can the findings of this study be applied to enhance the efficiency and environmental performance of solid oxide fuel cells?

Reviewer 3 Report

Comments and Suggestions for Authors

Table 1.1, Table 1.2 and Table 1.3 refer to the words 'Research' and 'Investigate'. It is not clear what do these terms refer to. Even in the text before/after the tables such terms need to be better explained.

It would be ideal if all chemical compounds are written in full prior to the first time of mentioning e.g. Methane (CH4), Particulate Matter (PM2.5)

It is not clear what is the involvement of the producers of the SOFC in the research. Is it just in providing the information on the SOFC, or was there any financial contribution to the research?

If none, then there certain paragraphs, such as production capability  and how many employees does the manufacturer have is superfluous (Lines 67-68).

Comments on the Quality of English Language

English has to be proof read, as the structure needs to be improved.
